# Intraguild Interactions between the Mealybug Predators *Cryptolaemus montrouzieri* and *Chrysoperla carnea*

**DOI:** 10.3390/insects12070655

**Published:** 2021-07-19

**Authors:** Laura Golsteyn, Hana Mertens, Joachim Audenaert, Ruth Verhoeven, Bruno Gobin, Patrick De Clercq

**Affiliations:** 1Department of Plants and Crops, Ghent University, Coupure Links 653, B-9000 Ghent, Belgium; laura.golsteyn@ugent.be (L.G.); hana.mertens@hotmail.com (H.M.); 2PCS—Ornamental Plant Research, Schaessestraat 18, B-9070 Destelbergen, Belgium; joachim.audenaert@pcsierteelt.be (J.A.); ruth.verhoeven@pcsierteelt.be (R.V.); bruno.gobin@pcsierteelt.be (B.G.)

**Keywords:** mealybug, greenhouse crops, biological control, predator, intraguild predation

## Abstract

**Simple Summary:**

The ladybird *Cryptolaemus montrouzieri* is a widely commercialized biological control agent of mealybugs. The green lacewing *Chrysoperla carnea* is mainly released for aphid control, but also attacks mealybugs. Both species have shown potential to control various economically important species of mealybug pests of greenhouse crops. As these predators may be simultaneously present in a crop, the risk of negative interactions between both predators was evaluated in this laboratory study. Individuals of different life stages of either predator were placed together in petri dish arenas and predation was recorded. Attacks between individuals of both species were frequently observed, with lacewing larvae being the dominant predators in most combinations. When mealybug nymphs or lepidopteran eggs were added to the arena, the incidence of attacks between the predators was greatly diminished. The relevance of these observations for the use of the predators in the biological control of greenhouse pests is discussed.

**Abstract:**

The ladybird *Cryptolaemus montrouzieri* and the green lacewing *Chrysoperla carnea* have shown potential for use in augmentative biological control of mealybug pests in greenhouse crops. In the context of combining these predators within an integrated pest management system, the risk of negative intraguild interactions between both predators was evaluated in a laboratory setting. Different life stages of either predator were confronted in petri dish arenas containing a *Ficus benjamina* leaf, and after 24 h the incidence and direction of intraguild predation (IGP) was recorded for each combination. The effect of adding *Planococcus citri* nymphs or *Ephestia kuehniella* eggs as extraguild prey on the level of IGP was also studied. IGP was frequently observed between the two predator species and was asymmetrical in favour of *C. carnea* in most cases. The presence of extraguild prey reduced the number of IGP events between the predators to a similar extent. The relevance of the observed intraguild interactions for the combined use of these predators in protected cultivation is discussed.

## 1. Introduction

Mealybugs are economically important pests in many greenhouse crops of temperate climates, including ornamentals [1,2]. Developing an efficient strategy to control mealybugs is challenging, particularly if an existing integrated pest management (IPM) system needs to be taken into consideration. Growers often resort to chemical pesticides to suppress mealybug outbreaks, but this interferes with the biological control of other pests. The ladybird *Cryptolaemus montrouzieri* (Mulsant) (Coleoptera: Coccinellidae) and the green lacewing *Chrysoperla carnea* (Stephens) (Neuroptera: Chrysopidae) are both predators with a proven ability to control mealybug pests [2,3,4,5]. *Chrysoperla carnea* is a highly generalistic predator with a wide climatic adaptability [6] and is currently mainly released for the control of aphids, whereas *C. montrouzieri* is a more thermophilic predator with a preference for mealybugs [7]. The simultaneous use of predators from the same guild in a biological control program may lead to trophic interactions among the different life stages of the species involved. Intraguild predation (IGP) may eventually affect the success of pest suppression [8,9]. *C. montrouzieri* has been reported to act as an intraguild predator of a number of natural enemies, mealybug parasitoids in particular [10,11,12], but also other predators [13,14]. The incidence of IGP between green lacewings and other predators in the aphidophagous guild has also been frequently documented, e.g., [15,16,17,18,19]. Several factors play a role in IGP incidence, intensity, direction, and symmetry. Ontogeny and size, which are correlated, together with the degree of feeding specificity and mobility, are among important factors that determine the outcome of IGP [20,21].

In the present study, stage-specific intraguild interactions between *C. montrouzieri* and *C. carnea* were evaluated in a laboratory setting, both in the absence and presence of extraguild prey (EGP). In the latter case, nymphs of the citrus mealybug, *Planococcus citri* (Risso) and eggs of the Mediterranean flour moth, *Ephestia kuehniella* Zeller (Lepidoptera: Pyralidae) were offered as extraguild prey. Citrus mealybugs are a natural/realistic prey for both predators, whereas *E. kuehniella* eggs were selected as they constitute a highly nutritional factitious food for both predators [22,23], used both for mass rearing and as a supplemental food in protected cultivation [2,24]. In contrast to *P. citri*, which are mobile and can display defence responses, these lepidopteran eggs are sessile and thus more easily accessible as prey.

## 2. Materials and Methods

### 2.1. Insect Rearing

Larvae of *C. carnea* were reared on *E. kuehniella* eggs in petri dishes (5.5 cm diameter). Adults of *C. carnea* were also kept in petri dishes (15 cm diameter) and fed pulverized honeybee pollen and honey diluted in water to which dry yeast was added. The *C. montrouzieri* colony was maintained according to the semi-artificial rearing system developed by [22], using *E. kuehniella* eggs as a food source and synthetic wadding as an artificial oviposition substrate. Colonies of both insects were kept in climatic chambers set at 25 ± 1 °C, 65 ± 5% RH, and a 16:8 (L:D) h photoperiod. *P. citri* was reared on potato sprouts and kept in complete darkness at the above-mentioned conditions of temperature and humidity. Founding individuals of both the *C. carnea* and *C. montrouzieri* cultures, in addition to frozen eggs of *E. kuehniella*, were provided by Koppert BV (Berkel en Rodenrijs, The Netherlands).

### 2.2. Intraguild Predation between Predator Larvae and Adults in the Absence of Extraguild Prey

Different combinations of the three larval instars of *C. carnea* and the four instars of *C. montrouzieri* were studied in individual confrontations. In addition, IGP of adult *C. montrouzieri* on *C. carnea* larvae was tested. *C. carnea* adults are not predatory and likely escape attacks by flying away so they were not considered in this study. In a vented petri dish (5.5 cm diameter, 1.5 cm height) containing an excised leaf of *Ficus benjamina*, a single individual of *C. carnea* was confronted with a single individual of *C. montrouzieri*. Each combination was replicated ten times. All predator immatures were newly (<24 h) emerged or moulted individuals taken from the laboratory colonies of either species. *C. montrouzieri* adults were ovipositing females, taken from the colony at emergence and given the chance for 10 to 15 days to mate and become reproductively active. The adults were starved for 24 h prior to the experiment. Individuals were weighed before the start of the experiment using a Sartorius Genius ME215P balance (Sartorius, Goettingen, Germany; precision: 0.1 mg). Initial body weights averaged 0.37 ± 0.05 (mean ± SE), 0.60 ± 0.10, and 1.93 ± 0.12 mg for first, second, and third instars of *C. carnea*, and 0.10 ± 0.01, 0.48 ± 0.05, 2.05 ± 0.19, 4.53 ± 0.21, and 12.91 ± 0.42 mg for first to fourth instars, and adults of *C. montrouzieri*, respectively. Each experiment lasted for 24 h, after which time the petri dishes were checked for mortality of either contestant to determine the incidence and direction of IGP. Individuals which fell victim to IGP could be identified by puncturing or biting injuries, or by the fact that they were partly or fully consumed, leaving only the cuticle [19]. In addition, ten control replicates were undertaken for each combination, in which predators were confined singly in a petri dish with a *F. benjamina* leaf, to check for natural mortality of the predators. All experiments were conducted in climatic cabinets set at 25 ± 1 °C, 65 ± 5% RH, and a 16:8 (L:D) h photoperiod.

### 2.3. Intraguild Predation on Immobile Predator Stages

IGP by *C. montrouzieri* larvae and adults on *C. carnea* eggs was also tested in 5.5 cm petri dishes containing a ficus leaf, as described above. One to two day old *C. carnea* eggs, deposited on stalks on a black piece of paper, were offered in groups of 10 eggs to a single *C. montrouzieri* individual. Completely consumed and partially damaged eggs were considered to be preyed upon and were counted. Predation by *C. carnea* larvae on *C. montrouzieri* eggs was not quantified in this study, because under natural conditions the ladybird deposits its eggs in the egg sacs of mealybugs. IGP on pupae of both species was also studied. For this purpose, the most recently formed pupae available in the culture were selected. IGP on pupae was estimated from the percentage of adult emergence adjusted for natural mortality, as attacks on pupae were not easy to ascertain. Ten pupae were maintained for 24 h under the same experimental conditions to determine the proportion of adult emergence in the absence of predators. Climatic conditions were as described above.

### 2.4. Intraguild Predation between Predator Larvae in the Presence of Extraguild Prey

Based on the results from the previous experiments, certain combinations of larval instars (Ln) of *C. carnea* versus *C. montrouzieri* (i.e., L1 vs. L1, L1 vs. L2, L1 vs. L3, L2 vs. L2, L2 vs. L3, L3 vs. L3, L3 vs. L4) were selected to study IGP in the presence of extraguild prey. An excess amount of *P. citri* nymphs (mixed second and third instars) or *E. kuehniella* eggs, adjusted to match the combined predation capacity of the confronted larval stages, was added to the petri dishes as EGP, but not replenished during the 24h experiment. The other experimental conditions were identical to those described for the tests without EGP.

### 2.5. Data Analysis

The experiments had three possible outcomes: no IGP occurred, *C. carnea* was the (successful) IGP predator, or *C. montrouzieri* was the IGP predator. How the outcome depended on the combined life stages of both predators was analysed by means of two-sided Fisher–Freeman–Halton exact tests on 2 × 3 tables. Exact tests were used to calculate p-values instead of χ^2^ tests, because of small sample sizes. The Benjamini–Hochberg method with an FDR of 0.05 was used to control for type I errors associated with multiple comparisons. The p-values were adjusted as described by [25]. To determine the effect of the presence and the type of EGP (no EGP, *P. citri* nymphs, or *E. kuehniella* eggs) on the outcome of IGP for each combination, two-sided Fisher–Freeman–Halton exact tests were performed on 3 × 3 tables. The Bonferroni correction was applied for post hoc multiple comparison testing.

The level of IGP (IL; %) was calculated for each pair of predators. This is the proportion of replicates with IGP over the total number of replicates [20]. In addition, the index of symmetry (SI; i.e., the number of replicates in which a certain predator was killed over the total number of replicates in which there was IGP) was determined [20]. An SI of 100% means that IGP is unidirectional in favour of *C. carnea*, whereas an SI of 0% defines a unidirectional IGP in favour of *C. montrouzieri*. In the case of mutual IGP, the index of symmetry was compared to a theoretical 50:50 distribution, corresponding to a completely symmetric interaction, using a goodness-of-fit exact test (for specific combinations) or χ^2^ test (for overall analysis).

A significance level of 5% was applied for all tests. SPSS Statistics was used to run the analyses.

## 3. Results

### 3.1. Intraguild Predation between Predator Larvae and Adults in the Absence of Extraguild Prey

Mortality in the controls never exceeded 10% within the 24 h test period, so treatment data were not corrected for natural mortality. The outcome of the experiments is shown in Table 1. Based on the observations, the IGP level (IL) was also calculated for each combination, in addition to the index of symmetry (SI).

When larvae of the same instars were confronted (L1 + L1, L2 + L2, L3 + L3), IGP occurred frequently and was unidirectional in favour of *C. carnea* larvae. Based on the initial weight of the predators involved in these confrontations, larvae of *C. carnea* were significantly heavier than their *C. montrouzieri* counterparts (*p* < 0.05, *t*-tests or Mann–Whitney U-tests).

Combining *C. carnea* larvae with larger *C. montrouzieri* individuals (i.e., larvae of older instars or adults) resulted in less fatal intraguild interactions for *C. montrouzieri*. For pairs with first and second instar larvae of *C. carnea,* IGP was no longer exclusively in favour of the lacewing, with *p* < 0.001 for contrasts (*C. carnea* vs. *C. montrouzieri*) L1 + L1 vs. L1 + L3, L1 + L1 vs. L1 + L4, L1 + L1 vs. L1 + adult, L1 + L2 vs. L1 + adult, L2 + L1 vs. L2 + adult, and L2 + L2 vs. L2 + adult, *p* = 0.0078 for contrast L2 + L1 vs. L2 + L3, and *p* = 0.026 for contrast L2 + L4 vs. L2 + adult (Fisher–Freeman–Halton exact tests, Table 1).

In the presence of third instars of *C. carnea*, *C. montrouzieri* larvae had no chance of surviving, irrespective of their size. Only in 10% of replicates, fourth instars of *C. montrouzieri* managed to survive when combined with third instars of *C. carnea*, which was statistically similar to the survival of younger ladybird instars (*p* > 0.05 in all cases).

*C. carnea* third instars could only be subdued by adults of *C. montrouzieri*. In confrontations between third instar lacewing larvae and ladybird adults, IGP was significantly reduced compared to the combinations with ladybird larvae and entirely in favour of *C. montrouzieri* (*p* < 0.001 for the contrasts L3 + L1, L3 + L2, L3 + L3, and L3 + L4 vs. L3 + adult) with the lowest number of fatal intraguild interactions occurring in combinations with third instars of *C. carnea*.

Although third and fourth instars of *C. montrouzieri* were heavier than second instars of *C. carnea* (*p* < 0.05, Mann–Whitney U-tests), they had similar or lower chances of being the IGP predator. The IGP outcome was similar to that in the combinations between second instars of *C. carnea* and second and third instars of *C. montrouzieri* (*p* > 0.05).

First instar larvae of *C. montrouzieri* were always preyed upon by *C. carnea*, irrespective of the latter’s larval stadium. Second instars of the ladybird were also mainly the prey in combinations with *C. carnea* larvae, irrespective of their instar (*p* > 0.05 for contrasts between the different instars of *C. carnea* when combined with second instars of *C. montrouzieri*). When third instars of *C. montrouzieri* were combined with first and second instars of *C. carnea*, the incidence of IGP was lower than when combining them with third instars of the lacewing (*p* < 0.01 for the corresponding contrasts). Combining fourth instars of *C. montrouzieri* with second and third instars of *C. carnea* significantly changed the IGP direction compared to combining the fourth instars of the ladybird with first instars of the lacewing (*p* < 0.05 for both contrasts). *C. montrouzieri* adults were IGP predators of all three *C. carnea* instars, but the level of predation was significantly higher on first instars of *C. carnea* than on third instars (*p* < 0.05).

IGP was only mutual for some of the combinations: none of the SI of the combinations between first instar *C. carnea* and second instar *C. montrouzieri,* and between second instars of the lacewing and third or fourth instars of the ladybird was significantly different from 50% (*p* > 0.05 in all cases). For all other combinations, IGP was unidirectional. Over all combinations, the SI was 72%, differing significantly from 50% (χ^2^ = 21.93, df = 1, *p* < 0.001). Thus, overall, the IGP between larvae or adults of *C. montrouzieri* and larvae of *C. carnea* was in favour of the latter.

### 3.2. Intraguild Predation on Immobile Predator Stages

First and second instar larvae of *C. montrouzieri* did not consume any *C. carnea* eggs within 24 h. IL-values when *C. montrouzieri* third instars, fourth instars, and adults were offered *C. carnea* eggs were 20%, 70%, and 100%, respectively, and 2%, 42%, and 98% of the offered batch of 10 eggs was consumed on average. Whereas in control replicates all *C. montrouzieri* pupae emerged, after attack by *C. carnea* L3 larvae, only 10% of replicates survived (*p* < 0.001). *C. montrouzieri* larvae nor adults attacked *C. carnea* pupae.

### 3.3. Intraguild Predation between Predator Larvae in the Presence of Extraguild Prey

The presence of *P. citri* nymphs in the arenas significantly changed the IGP outcome for all combinations as compared with combinations without EGP (L1 + L1: *p* < 0.001, L1 + L2: *p* = 0.011, L2 + L2: *p* = 0.001, L2 + L3: *p* = 0.011, L3 + L3: *p* = 0.003, L3 + L4: *p* < 0.001). Only for the L1 + L3 combination, the level of IGP, which was in favour of *C. montrouzieri*, was not significantly reduced by the presence of *P. citri* nymphs (*p* = 0.087).

For the combinations L2 + L2 (*p* < 0.001), L2 + L3 (*p* = 0.011), and L3 + L4 (*p* = 0.001), having access to *E. kuehniella* eggs significantly reduced the IGP predation by *C. carnea*, whereas for L3 + L3 predation by both *C. carnea* and *C. montrouzieri* was lower (*p* < 0.001). For the pairs involving first instars of *C. carnea*, the presence of *E. kuehniella* did not affect the incidence of IGP (L1 + L1: *p* = 0.087, L1 + L2: *p* = 0.475, L1 + L3: *p* = 0.087).

Overall, there were no significant differences regarding the outcome of IGP between offering *P. citri* nymphs or *E. kuehniella* eggs as EGP (*p* > 0.05 for all tested combinations).

In cases where IGP less commonly occurred for a pair of predators without EGP, the presence of *P. citri* mealybugs reduced the IGP level to zero. In combinations with high IGP levels, the presence of an EG prey reduced the IGP level, but IGP did not disappear completely.

IGP was only mutual for some of the combinations in the presence of EGP. None of the SI-values for the combinations with mutual IGP (i.e., L1 + L1 + *P. citri*, L1 + L2 + *E. kuehniella*, and L3 + L3 + *P. citri*) differed significantly from 50% (*p* = 0.219, *p* = 1, and *p* = 0.625, respectively). For all other pairs, IGP was unidirectional.

## 4. Discussion

Several mealybugs, including *P. citri, Pseudococcus longispinus* (Targioni Tozzetti), and *Pseudococcus viburni* (Signoret)*,* increasingly cause damage in a range of greenhouse ornamental and vegetable crops in temperate zones [1,2]. Chemical control of mealybugs is difficult due to their waxy cover and the development of pesticide resistance [26]. Moreover, repeated insecticide applications required to suppress mealybug outbreaks may disrupt biological control programs against other pests. As a consequence, the augmentative release of parasitoids and predators against mealybug pests has received increasing attention. Whereas parasitoids may be more effective against mealybug infestations by full field releases [27,28], predators such as the larvae of generalist lacewings, and the larvae and adults of the more specific mealybug predator *C. montrouzieri*, have potential for use in pest hot spots [2]. Given their generalist feeding, lacewing larvae may also be used to suppress other pests in the crop, such as aphids and spider mites [6,29]. *Cryptolaemus montrouzieri* and *C. carnea* may thus be simultaneously present in the crop, and as a consequence might engage in intraguild interactions. The interactions between these predators have not been previously investigated.

In the present laboratory study, IGP was observed in 76% of confrontations between larvae and adults of *C. montrouzieri* and larvae of *C. carnea*. IGP events were recorded in both directions. All tested *C. montrouzieri* stages, except adults, were vulnerable to lacewing larvae. All tested *C. carnea* stages were prone to attack by coccinellid adults and larvae, except pupae.

Sessile stages are expected to be more vulnerable to predation, particularly when tested in small scale arenas, unless they possess morphological or behavioural defences [20,30]. Despite being offered on their stalks, lacewing eggs were heavily preyed upon by older instars and adults of *C. montrouzieri*. Similarly, Ref. [20] reported that lacewing eggs were vulnerable to attack by the ladybird *Coleomegilla maculata lengi* Timberlake. In a study by [14], pupae of the coccinellid *Nephus includens* (Boheman) were preyed upon by *C. montrouzieri*. In contrast, the silk cocoon of *C. carnea* pupae may have offered this life stage protection from predation by the coccinellid in the present study [18]. Predation by *C. carnea* larvae on *C. montrouzieri* eggs was not quantified in this study, because in the field the ladybird deposits its eggs in or near egg sacs of mealybugs. In the study by [20], however, coccinellid egg masses were highly vulnerable to predation by lacewing larvae.

The experiments combining mobile stages of both predators indicate strong asymmetrical interactions for several of the tested combinations, with *C. carnea* being the dominant IGP predator. IGP occurred in 84% of the tests involving third instars of *C. carnea*. The interaction was always asymmetrical and favoured the third instar lacewing, except when paired with *C. montrouzieri* adults, which may have gained protection from their sclerotized integument [19]. Strong aggressiveness of third instar *C. carnea* was also noted against late instar larvae of the ladybirds *Coleomegilla maculata* (De Geer) [15], *Harmonia axyridis* (Pallas) [17], and *Hippodamia variegata* (Goeze) [19].

Ontogeny of interacting individuals is known to affect the outcome of IGP [8,21]. In the present study, there was also a significant effect of life stage and body size of both interacting species on the frequency and direction of IGP. Early instars of *C. montrouzieri* were most vulnerable to IGP, whereas older larvae of *C. carnea* and adults of *C. montrouzieri* were the strongest intraguild predators. In many of the tested combinations, the larger individuals behaved as intraguild predators on the smaller intraguild prey. This effect was clearest in the absence of EGP. However, second and third instars of *C. carnea* were able to subdue larger *C. montrouzieri* contestants in several combinations (e.g., L2 vs. L3, L2 vs. L4, L3 vs. L4). As in several other studies [30,31,32], ladybird larvae also lost the contest with lacewing larvae of similar size (e.g., L1 vs. L2, L3 vs. L3).

In addition to body size, behavioural or morphological attributes can affect the outcome of intraguild interactions [20,30,33,34]. Greater mobility may allow a predator to escape IGP [35]. In our study, *C. carnea* larvae were overall more agile than *C. montrouzieri* larvae. Further, the sickle-shaped mouthparts of lacewing larvae enable them to more effectively attack and feed on larval ladybird prey [19,30]. Larvae of *C. montrouzieri* are covered by waxy filaments, which can serve as protection against potential predators [36,37]. However, our findings suggest that this type of protection may not be very effective against lacewing larvae. The study by [17] also indicated that the parascoli with hairs and alkaloids of *H. variegata* larvae were ineffective as a defence against lacewing larvae. The white waxy filaments are also a mimicry strategy of *C. montrouzieri* larvae. The larvae resemble mealybugs and are therefore camouflaged in colonies of their mealybug prey. This could be a partial explanation for why there was less IGP on larvae of the ladybird in the presence of *P. citri*.

Adding extraguild prey ad libitum to the experimental arena resulted in a decrease in the level of IGP, although it was still observed for some combinations (e.g., L1 vs. L1 with eggs of *E. kuehniella,* L3 vs. L3 with *P. citri* nymphs). A decrease in levels of IGP in the presence of extraguild prey can be explained by several mechanisms. First, this may be the result of differences in quality between the intraguild and extraguild prey. This can be related to a higher nutritional value of the extraguild prey, or lower energy expenses and lower risk of injury when attacking extraguild prey as compared with attacking another predator [38]. Shifting focus to extraguild prey may further reduce chances of encounter among predators, even in small arenas, and thus lead to a lower level of intraguild interactions [33,38]. Finally, easy access to nutritionally adequate extraguild prey may improve vigour of an intraguild prey and enhance its defence capabilities against an intraguild predator [39].

Overall, the effect of the presence of *E. kuehniella* eggs on IGP was not different from that of *P. citri* nymphs. Eggs of *E. kuehniella* have been noted to be an adequate factitious food for different insect predators, including *C. montrouzieri* and *C. carnea,* and are in many cases nutritionally superior to their natural prey [22,23,40]. In addition, as these lepidopteran eggs are sessile, the predators have no energy expenditures for searching and subduing this type of prey. Together, this would be expected to lead to a higher level of satiation than when having access to mealybugs and consequently result in a faster decrease in intraguild interactions between the predators. On the other hand, the lower nutritional quality of mealybugs may have been compensated in part by their mobility and thus greater attractiveness to the predators [41]. Consumption of extraguild prey by either predator was not quantified in this study.

It is worth noting that, despite the fact that natural mortality in the control treatments never exceeded 10%, predation data reported in Table 1 are likely the combined effect of direct IGP and exhaustion due to escape behaviours and starvation. This may have led to an overestimation of IGP levels in certain combinations. Further, habitat complexity may affect intensity, symmetry, and direction of IGP between insect predators [38,42,43]. In the present study, the interactions between *C. carnea* and *C. montrouzieri* were studied in small petri dish arenas offering few refuges. Such small artificial arenas merely serve to show the potential outcome of predatory interactions between two species [44]. Experiments on plants may provide a more reliable prediction of the incidence of IGP between the studied predators in the field. In a number of previous studies, the size and structural complexity of the arena had a clear effect on the intensity of IGP, with smaller and more simple arenas generating higher levels of IGP [38]. For instance, [45] found that, although *Chrysoperla plorabunda* (Fitch) and *Coccinella septempunctata* L. larvae consumed each other in petri dishes, they did not affect one another’s impact on aphid populations on potted broad bean plants. In the experiments of [33], IGP still occurred between *C. carnea*, *C. septempunctata*, and *Episyrphus balteatus* De Geer on broad bean plants, but it was reduced by approximately fivefold compared with that in petri dishes. Conversely, other studies reported less substantial effects of arena complexity on IGP levels, e.g., [19,41]. 

In conclusion, the present laboratory study indicates frequent IGP between different stages of the mealybug predators *C. carnea* and *C. montrouzieri*, and this was asymmetrical in favour of *C. carnea* in most cases. These intraguild interactions should therefore be taken into account when using both species together in biological control programs. Arguably, it is difficult to extrapolate the results from the present petri dish experiments to predict the incidence of IGP between *C. carnea* and *C. montrouzieri* in the field. Greenhouse studies with different host plants and varying levels of extraguild prey (both focal prey and supplemental foods) are needed to improve our understanding of the interactions among both predators in protected cultivation. Molecular tools can also assist in assessing the ecological relevance of IGP in greenhouse cropping systems [46].

## Figures and Tables

**Table 1 insects-12-00655-t001:** Outcome of intraguild predation (IGP), IGP level (IL), and index of symmetry (SI) for combinations of life stages of *C. carnea* and *C. montrouzieri* without extraguild prey, with *E. kuehniella* eggs as extraguild prey, or with *P. citri* nymphs as extraguild prey.

Combination of *C. carnea* + *C. montrouzieri*	Extraguild Prey	*C. carnea* as IGP Predator (%)	*C. montrouzieri* as IGP Predator (%)	IL (%)	SI (%)
L1 + L1	None	100	0	100	100
	*E. kuehniella* eggs	60	0	60	100
	*P. citri* nymphs	10	10	20	50
L1 + L2	None	50	10	60	83
	*E. kuehniella* eggs	20	10	30	67
	*P. citri* nymphs	0	0	0	-
L1 + L3	None	0	40	40	0
	*E. kuehniella* eggs	0	0	0	-
	*P. citri* nymphs	0	0	0	-
L1 + L4	None	0	50	50	0
L1 + adult	None	0	100	100	0
L2 + L1	None	100	0	100	100
L2 + L2	None	90	0	90	100
	*E. kuehniella* eggs	0	10	10	0
	*P. citri* nymphs	10	0	10	100
L2 + L3	None	30	30	60	50
	*E. kuehniella* eggs	0	0	0	-
	*P. citri* nymphs	0	0	0	-
L2 + L4	None	60	10	70	86
L2 + adult	None	0	50	50	0
L3 + L1	None	100	0	100	100
L3 + L2	None	100	0	100	100
L3 + L3	None	100	0	100	100
	*E. kuehniella* eggs	0	10	10	0
	*P. citri* nymphs	30	10	40	75
L3 + L4	None	90	0	90	100
	*E. kuehniella* eggs	10	0	10	100
	*P. citri* nymphs	0	0	0	-
L3 + adult	None	0	30	30	0

## Data Availability

The data presented in this study are available on request from the corresponding author. The data are not publicly available due to ownership by the funders.

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
