# Peer review of "Intraguild Interactions between the Mealybug Predators Cryptolaemus montrouzieri and Chrysoperla carnea"

_insects, 2021, doi:10.3390/insects12070655_

Round 1
Reviewer 1 Report
I would say that the submitted paper is potentially acceptable for the publication of Insects because the topic described in the paper is interesting for the field of applied entomology. I would feel that the research was generally well done.
The following points must be cleared before the acceptance of Insects.
Major point
Please explain more precisely the criterion in M & M how you decide to the larva of the ladybird or the green lacewing was dead with IGP (i.e., the death may be caused by IGP or starvation, especially for no food condition. How do you distinguish among them?).
I would think that the percentage of IGP (C. carnea or C. montrouzieri) in your paper (Table 1) may reflect the combined effects of intensity of IGP, starvation, or escaping behaviors: There is a possibility that the survived larva with higher starvation and/or mobile ability, may be a better survivor than less one even if no IGP occurs. In this case, the survivor cannot be regarded as IGP predator.
This point is very important for the reliability of a key result in your paper.
Minor point
L68-71 Please explain the origin of the population of C. carnea and C. montrouzieri with sample size (i.e., you collected from a field or obtained from companies et al.).
Author Response
Major point
Please explain more precisely the criterion in M & M how you decide to the larva of the ladybird or the green lacewing was dead with IGP (i.e., the death may be caused by IGP or starvation, especially for no food condition. How do you distinguish among them)?
Response: This is a good point and an overlook from our side. We have basically used the same methodology as Zarei et al. (Insects 2020, 11, 11, doi:10.3390/insects11050288) who looked at signs of predation or evidence of consumption in their experiments. In addition, we also ran control treatments for all combinations allowing us to estimate natural mortality. We have specified this in the revised paper: "Individuals which fell victim to IGP could be identified by puncturing or biting injuries or by the fact that they were partly or fully consumed, leaving only the cuticle [19]. In addition, ten control replicates were done for each combination, in which predators were confined singly in a petri dish with a F. benjamina leaf, to check for natural mortality of the predators."
I would think that the percentage of IGP (C. carnea or C. montrouzieri) in your paper (Table 1) may reflect the combined effects of intensity of IGP, starvation, or escaping behaviors: There is a possibility that the survived larva with higher starvation and/or mobile ability, may be a better survivor than less one even if no IGP occurs. In this case, the survivor cannot be regarded as IGP predator.
Response: We thank the reviewer for pointing this out as this is indeed another valid point. To accomodate this comment, in the section of the Discussion highlighting the limitations of our study we have now added: "It is worth noting that, despite the fact that natural mortality in the control treatments never exceeded 10%, predation data reported in Table 1 are likely the combined effect of direct IGP and exhaustion due to escape behaviors and starvation. This may have led to an overestimation of IGP levels in certain combinations. Further, habitat complexity may affect intensity, symmetry and direction of IGP between insect predators [38,42,43]...". Indeed, one might judge that individuals which died as a result of exhaustion due to being chased by a potential predator are in fact also a victim of intraguild predation - even when they are not eaten. Finally, we wish to add that this type of widely used experimental setup for assessing IGP does not allow to discriminate whether an individual gets eaten after it has been killed by a direct successful attack from a predator or it gets eaten after it had already died of other causes (necrophagy).
Minor point
L68-71 Please explain the origin of the population of C. carnea and C. montrouzieri with sample size (i.e., you collected from a field or obtained from companies et al.).
Response: We have included this information in the revised paper: "Founding individuals of both the C. carnea and C. montrouzieri cultures, as well as frozen eggs of E. kuehniella were provided by Koppert BV (Berkel en Rodenrijs, The Netherlands)."
Reviewer 2 Report
The paper is well written and the topic is interesting even if the results are bringing relative new informations. Indeed, larger size predator have the advantage for IGP and external preys in the environment decrease IGP. Nevertheless, the data's are bringing one more illustration of this topic.
One table to present all the results mixing all experiments is not enough and clear. One figure by experiment with a more visual way to present is needed and should be proposed.
Minor comments:
- line 93: "for first to fourth instars" should replace "first, second, third, fourth",
- statistical values are sometimes only probabilities and sometimes p value with khi2 values. The latter have to be added to homogenize the presentation.
Author Response
The paper is well written and the topic is interesting even if the results are bringing relative new informations. Indeed, larger size predator have the advantage for IGP and external preys in the environment decrease IGP. Nevertheless, the data's are bringing one more illustration of this topic.
Response: We appreciate the positive comments by the reviewer.
One table to present all the results mixing all experiments is not enough and clear. One figure by experiment with a more visual way to present is needed and should be proposed.
Response: We have explored the possibilities for a visual presentation of the data but the large number of combinations studied complicates a simple graphic presentation that would be clearer than and as complete as the current Table 1. We regret to say that we do not fully understand what the reviewer means by "one figure by experiment". There are basically 30 experiments (15 combinations of life stages with or without extraguild prey), so if we read the comment well we would need at least 15 figures. Looking at our dataset in Table 1, we could perhaps select out the IL values for separate presentation in a graph, but we doubt that a bar graph with 30 bars and values ranging from 0 to 100 is going to make the data clearer for the reader. We do not see the point either in selecting only certain combinations for visual presentation, as this would complicate comparison of the different treatments. We would very much welcome a concrete suggestion from the reviewer on which type of graph would be suitable to present (a coherent part of) the data in Table 1.
Minor comments:
- line 93: "for first to fourth instars" should replace "first, second, third, fourth"
Response: Done
- statistical values are sometimes only probabilities and sometimes p value with khi2 values. The latter have to be added to homogenize the presentation.
Response: We understand that the text may have caused some confusion. We have only used χ2 tests if the dataset was sufficiently large, i.e. for the overall analysis of SI-values in the experiments without extraguild prey. Hence, the χ2 value in line 199. In all other cases, Fisher-Freeman-Haltonexact tests were used, so there was no χ2 value, only a p-value. This is now better explained in the Data Analysis section: "How the outcome depended on the combined life stages of both predators was analyzed by means of two-sided Fisher-Freeman-Halton exact tests on 2x3 tables. Exact tests were used to calculate p-values instead of χ2 tests, because of small sample sizes." and further "In the case of mutual IGP, the index of symmetry was compared to a theoretical 50:50 distribution, corresponding to a completely symmetric interaction, using a goodness-of-fit exact test (for specific combinations) or χ2 test (for overall analysis)."
Round 2
Reviewer 1 Report
The mentioned points are better revised.